# UP-VLA: A Unified Understanding and Prediction Model for Embodied Agent

Jianke Zhang [* 1]   Yanjiang Guo [* 1 2]   Yucheng Hu [* 1]   Xiaoyu Chen [1 2]   Xiang Zhu [1]   Jianyu Chen [1 2]

## Abstract

Recent advancements in Vision-Language-Action (VLA) models have leveraged pre-trained Vision-Language Models (VLMs) to improve the generalization capabilities. VLMs, typically pre-trained on vision-language understanding tasks, provide rich semantic knowledge and reasoning abilities. However, prior research has shown that VLMs often focus on high-level semantic content and neglect low-level features, limiting their ability to capture detailed visual and spatial information. These aspects, which are crucial for robotic control tasks, remain underexplored in existing pre-training paradigms. In this paper, we investigate the training paradigm for VLAs, and introduce **UP-VLA**, a **U**nified VLA model training with both multi-modal **U**nderstanding and future **P**rediction objectives, enhancing both high-level semantic comprehension and low-level spatial understanding. Experimental results show that UP-VLA achieves a 33% improvement on the Calvin ABC-D benchmark compared to the previous state-of-the-art method. Additionally, UP-VLA demonstrates improved success rates in real-world manipulation tasks, particularly those requiring precise spatial information. Code can be found at https://github.com/CladernyJorn/UP-VLA.

## 1. Introduction

Constructing Vision-Language-Action (VLA) models (Brohan et al., 2023; Li et al., 2023b) capable of solving multiple tasks in open environments has become a central focus of research in robotics. A promising approach for VLA models involves fine-tuning large-scale pre-trained Vision-Language Models (VLMs) (Li et al., 2023a; Wang et al.,

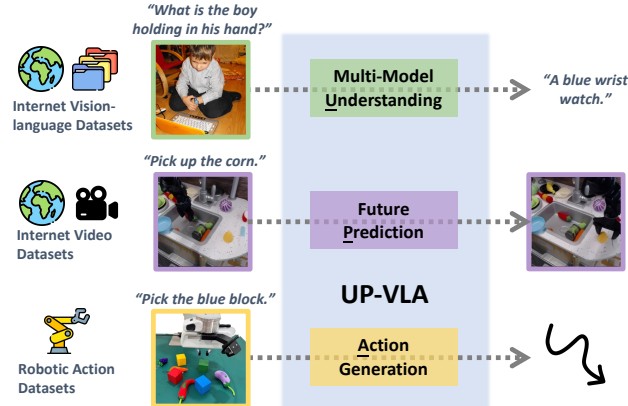

*Figure 1.* UP-VLA is pre-trained with both multi-modal understanding objective and future prediction objective to better capture both high-level semantic information and low-level spatial details, enhancing embodied decision-making tasks.

2022; Dai et al., 2024; Driess et al., 2023) on robotic action datasets, incorporating appropriate action modeling components (Jang et al., 2022; Li et al., 2023b; Wu et al., 2023; Zhang et al., 2024; Kim et al., 2024; Zheng et al., 2024b; Cui et al., 2025). This method enables VLA models to inherit the semantic knowledge and reasoning capabilities encoded in powerful VLMs, thereby enhancing decision-making in unknown environments.

However, previous works have identified certain weaknesses in VLMs, particularly in capturing low-level information and understanding physical dynamics (Zheng et al., 2024a; Chen et al., 2024a). Zheng et al. (2024a) highlighted that VLMs are weak in low-level vision tasks without additional training. Chen et al. (2024a); Wen et al. (2024) pointed out that pretrained VLMs lack spatial understanding and fail to capture low-level details such as distance and size differences. Furthermore, studies (Balazadeh et al., 2024; Ghaffari & Krishnaswamy, 2024; Li et al., 2024) have revealed significant challenges in VLMs' ability to understand physical dynamics. These limitations are largely attributed to the pre-training paradigm of VLMs (Wen et al., 2024; Chen et al., 2024a), which prioritizes multi-modal understanding tasks, such as Visual Question Answering (VQA), that enhance semantic reasoning but may overlook low-level details that are crucial for embodied decision-making tasks. While the generalization advantages offered by current pre-

*Equal contribution [1]Institute for Interdisciplinary Information Sciences, Tsinghua University, Beijing, China. [2]Shanghai Qi Zhi Institute, Shanghai, China. Correspondence to: Jianyu Chen <jianyuchen@tsinghua.edu.cn>.

*Proceedings of the 42nd International Conference on Machine Learning*, Vancouver, Canada. PMLR 267, 2025. Copyright 2025 by the author(s).

training approaches are desirable, they raise an important question: can a better training pipeline be developed to combine the strengths of both worlds, retaining semantic understanding while also emphasizing low-level features critical for control?

In this paper, we re-consider the pre-training approach for VLA models. Rather than focusing exclusively on high-level semantic information like in vision-language pre-training, we propose a training paradigm that emphasizes both high-level semantic understanding and low-level visual patterns. Inspired by prior papers on visual pre-training (Wu et al., 2023; Guo et al., 2024), we introduce a novel training paradigm for VLA models that aligns representations with both high-level features using the multi-modal understanding dataset and low-level features through future predictive generation. Specifically, we co-train an autoregressive model with a flexible attention mask on three types of datasets, as illustrated in Figure 1.

Our experiments validate the effectiveness of our new training paradigm for VLA models. As summarized in Figure 2, we tested three clusters of tasks ranging from simulation to real world settings to assess different abilities of the algorithms. ABCD→D and real-seen focus on evaluating the model's in-distribution multitask learning capabilities, real-unseen focus on real-world semantic generalization, while the remaining two tasks measure the methods' adaptation and precise control abilities. In alignment with our previous analysis, the VLM-based VLA model demonstrates relatively strong performance in both in-distribution multitask settings and the real-world generalization task (real-unseen). Conversely, visual prediction-based pre-training achieves relatively better scores in tasks requiring adaptation and precise control (real-precise and ABC-D). Notably, UP-VLA achieves a 33% improvement on the Calvin ABC→D generalization benchmark and shows significant improvement in real-world task. These results highlight the effectiveness of UP-VLA method in retaining both semantic and low-level features. Our contributions can be summarized as follows.

1. Motivated by recent insights into the limitations of VLMs, we integrate video datasets rich in detailed information and dynamic contexts into the pre-training of VLA models to enhance their capabilities.

2. We introduce a novel training paradigm for VLA models that combines both vision-language understanding and future prediction objectives, enabling the capture of both high-level semantic and low-level visual patterns essential for embodied agents.

3. We achieve significant improvements in success rates across both simulated and real-world manipulation tasks. Additionally, we conduct an ablation study to validate the effectiveness of two types of pre-training.

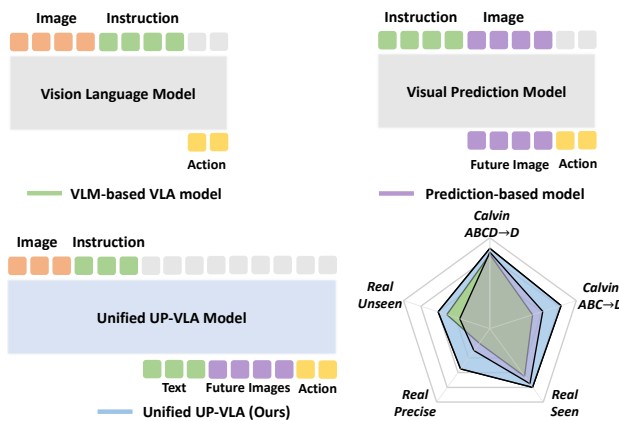

*Figure 2.* Comparison between UP-VLA, VLM-based VLA models and prediction-based models. The bottom-right chart illustrates the performance across multiple tasks in both simulated and real-world environments. We select the best model from each type of method

## 2. Related Works

**VLA Models for Generalist Robot Policies** Recent studies have explored the application of VLMs (Li et al., 2023a; Wang et al., 2022; Dai et al., 2024; Driess et al., 2023; Wang et al., 2023; Zhao et al., 2025; Ding et al., 2025; Song et al., 2025; Ding et al., 2024) in robotics, leveraging their strong understanding of linguistic instructions and visual scenes. A notable example is RT-2 (Brohan et al., 2023), which directly utilizes VLMs to generate discrete action tokens autoregressively, demonstrating the semantic grounding capabilities of VLA methods. Recent works aim to enhance VLA models with better generalization performance (Kim et al., 2024; O'Neill et al., 2023), cross-embodiment control capabilities (Black et al., 2024), and improved reasoning efficiency (Zhang et al., 2024). The previous work, 3D-VLA (Zhen et al., 2024), also explored co-training for multi-modal understanding and generation, but focused on introducing 3d information and employed a separate diffusion model for generation. In contrast, our approach uses a unified model to handle multi-modal input and mainly focuses on addressing the limitations of VLA models in visual perception and physical dynamics.

**Visual Pretraining Methods for Robotics** Leveraging pre-trained vision models for robotic perception has become a crucial area of research in robot control. Early works (Brohan et al., 2022; Jang et al., 2022) employed pretrained vision encoders like ViT (Dosovitskiy et al., 2020) and EfficientNet (Tan & Le, 2019) to encode visual observations. Recently, numerous studies have incorporated generative models (Ho et al., 2020; Blattmann et al., 2023) for training policies via future frame prediction (Guo et al., 2024; Ding et al., 2024) and video generation (Du et al., 2024). For example, SuSIE (Black et al., 2023) learns robot actions by predicting keyframes, while GR-1 (Wu et al., 2023)

directly pretrains policies through video generation. PAD (Guo et al., 2024) employs diffusion models to predict future images and multi-step actions simultaneously. IGOR (Chen et al., 2024b) uses latent actions that compress visual changes as the intermediate goal for low-level actions. These studies highlight that visual prediction tasks can benefit models' visual generalization to unseen scenes. Our approach leverages autoregressive VLMs to predict future images, capturing physical dynamics with rich visual information.

## 3. Preliminaries

**VLA for Language Conditioned Robot Control** The language-conditioned manipulation problem is considered a decision sequence under the environment modeled by a free-form language instruction $l$ specifying a certain task and the initial observation $o_1$. For demonstrations $\mathcal{D} = \{\tau_1, \tau_2, \cdots, \tau_n\}$, where each frame $\tau_i = \{(o_t, a_t)\}_{t=1}^T$ consists visual observations $o$ and actions $a$. Vision-Language-Action (VLA) models typically train VLM$\pi_\theta$ as a robotic action policy by minimizing the error between $\hat{a} \sim \pi_\theta(o, l)$. Leveraging the multi-modal understanding capability of VLM, VLA has better generalization across tasks, especially enhanced semantic understanding of unseen objects and improved ability to understand or reason complex natural language instructions.

**Unified Vision-Language Pretraining via Autoregressive Modelling** Unified multi-modal language models are capable of understanding and generation. An effective and scalable approach is to improve the VLMs with an additional image generation task, as demonstrated in works like SeeD-X (Ge et al., 2024) and Showo (Xie et al., 2024). Following these approaches, we utilize a discrete image encoder to handle encoding and decoding for image-generation tasks, while employing a continuous vision encoder for multi-modal understanding and reasoning tasks. During training, the LLM input is prompted based on the task type in the following format:

$$\{|MMU|, (u_1, u_2, \cdots, u_n), (l_1, l_2, \cdots, l_m)\}$$
$$\{|T2I|, (l_1, l_2, \cdots, l_m), (v_1, v_2, \cdots, v_n)\}$$

where $l$ represents language tokens, and $u$, $v$ correspond to continuous and discrete image tokens for different tasks.

## 4. Methodology

Our goal is to develop a better training schema for VLAs. In this section, we describe the details of UP-VLA. We first build our backbone on top of a unified VLM. Then, we design a unified mechanism to bridge the gap between visual prediction with multi-modal understanding. Finally, we enhance action learning with unified prediction and un-

derstanding prompting techniques.

### 4.1. Backbone

As illustrated in Figure 3, we employ Phi-1.5 (Li et al., 2023c) as the underlying large language model. For multi-modal understanding tasks, we follow the standard VLM encoding approach, projecting images into the language embedding space via a CLIP-ViT (Radford et al., 2021) encoder. These projected image features are then concatenated with language embeddings and fed into the large language model. For image prediction tasks, we encode the currently observed image into discrete tokens using VQ-GAN (Esser et al., 2021). Instead of using noise prediction or masked reconstruction, we directly predict future image tokens in the corresponding position of output tokens, which encourages the model to focus on the visual information in the current frame and predict future changes conditioned on language.

### 4.2. Bridging Visual Prediction and Multi-modal Understanding

To enable LLMs to possess both visual prediction and multi-modal understanding capabilities, we incorporate both future prediction tasks from robotics data and image-text pairs during training. These two types of tasks can be encoded into a unified format so they can be mixed and processed in parallel through the LLM backbone. Therefore, we extend the multitasking approach described in sec 3.

**Multi-modal Understanding** Given a paired image-text question-answering set $(I, L)$, we encode the image into the language embedding space via a continuous encoder and a connection layer $E_1$, resulting $\mathbf{u} = \{u_i\}_{i=0}^M = E_1(I)$. These embeddings are concatenated with text embeddings $\mathbf{l} = \{l_i\}_{i=0}^N$ to form the multi-modal input. To generate a text sequence that can focus on the image while comprehending the language, we modify the causal attention mechanism so that image tokens can attend to each other, as shown in Figure 4(a). Finally, we use an autoregressive manner to predict the next language token. This task can be briefly described as $\hat{L} = \pi_\theta^{MMU}(I, L)$.

**Future Visual Prediction** For image prediction, given an image and instruction pair $(O_t, L)$ at time $t$, we encode the current visual observation using a discrete encoder $E_2$: $\mathbf{v}_t = \{v_i\}_{i=0}^M = E_2(O_t)$. Unlike multi-modal understanding tasks, the objective in visual prediction is to encode future visual observation by focusing on the instruction prompts. Thus, as depicted in Figure 4(b), we place the image tokens after the language tokens, enabling the image to attend to all input information. Meanwhile, we introduce a new special token $PRE$ to denote this new task. Instead of using next token prediction, we model the future image tokens at the

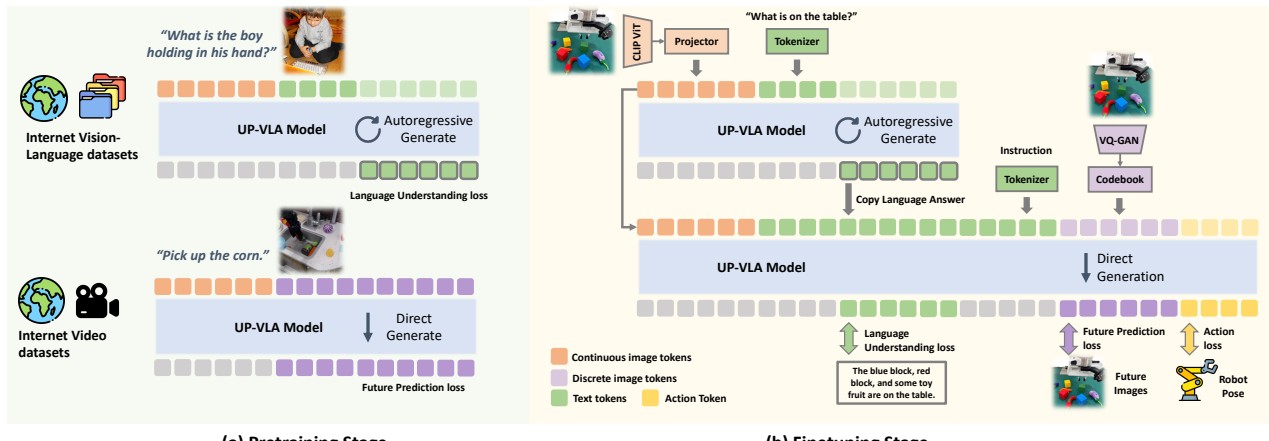

*Figure 3.* Overview of UP-VLA. Our model unifies visual-language understanding, future image generation, and action learning in an autoregressive manner. It takes the current visual scene and language instructions as inputs, produces a high-level understanding of the scene, and subsequently predicts future images and robotic actions based on these understanding tokens.

same positions of the image tokens:

$$P(O_{t+\Delta t}|O_t, L) = \prod_{i=1}^{M} p_\theta(v_{t+\Delta t}^i|\mathbf{v}_t, l)$$

, and then use a discrete decoder to reconstruct the predicted future observation image $\hat{O}_{t+\Delta t}$. This task can be succinctly described as $\hat{O}_{t+\Delta t} = \pi_\theta^{PRE}(O_t, L)$.

### 4.3. Enhancing Action Learning with Joint Prediction and Understanding

While previous VLA method leverages the muli-modal understanding knowledge of pre-trained VLM, it fails to exploit the rich visual information and the physical dynamics. To address this limitation, we propose a joint prediction-and-understanding action learning mechanism. We integrate action output with image prediction tasks. Given the current observation-instruction pair $(O_t, L)$, our model predicts both future observations and a sequence of actions at each time step: $(\hat{O}_{t+\Delta t}, \hat{A}_{t:t+\Delta t}) = \pi_\theta^{PRE}(O_t, L)$, where $\hat{A}$ corresponds to the final layer features at the positions of the action tokens.

In addition, as shown in Figure 4(c), we extend the language instruction input with scene descriptions generated by the model itself. The expanded input prompt is:

$$L' = [E_1(O_t'), \pi_\theta^{MMU}(O_t, L_{\text{prompt}}), L]$$

where $L$ is the language instruction and $O_t'$ represents various visual information at the current time step. The observation $O_t'$, after processing through the continuous vision encoder $E_1 = MLP(VIT)$, is mapped into the language embedding space $E_1(O_t')$ which can be directly used as language tokens. The final component $\pi_\theta^{MMU}(O_t, L_{\text{prompt}})$ is the generated description of the current scene, where $L_{\text{prompt}}$ is a specific prompt, such as "describe this image".

Finally, we generate actions via joint prediction:

$$(\hat{O}_{t+\Delta t}, \hat{A}_{t:t+\Delta t}) = \pi_\theta^{PRE}(O_t, L')$$

We use a small policy head to output low-level actions, consisting of a MAP module (a single-layer attention module) and a linear layer: $\hat{a}_{t:t+\Delta t} = MLP(MAP(\hat{A}_{t:t+\Delta t}))$

### 4.4. Training Strategy

We initialize the backbone of UP-VLA using Show-o (Xie et al., 2024). During training, we fully fine-tune the parameters of the LLM and freeze all encoders. The training process can be divided into two stages. In the first stage, we aim to endow the VLM with both visual prediction and multi-modal understanding capabilities. In the second stage, we focus on learning actions using robot data. We apply different sampling ratios for different tasks.

#### 4.4.1. TRAINING PIPELINE

**Prediction and Understanding Pretraining Stage.** We mix training data across two domains: one part is from Bridge (Walke et al., 2023), which includes 25k robotic arm demonstrations. We use this data for future prediction. Another part is from LLava-tuning-665k (Liu et al., 2024), which includes 665k image-text pairs for enhancing high-level understanding capability.

**Prediction with Action Tuning Stage.** The model is fine-tuned on downstream embodied tasks. We train UP-VLA using the joint prediction-and-understanding action learning approach in sec 4.3. We continue to co-train with the image-text pairs to preserve multi-modal understanding ability.

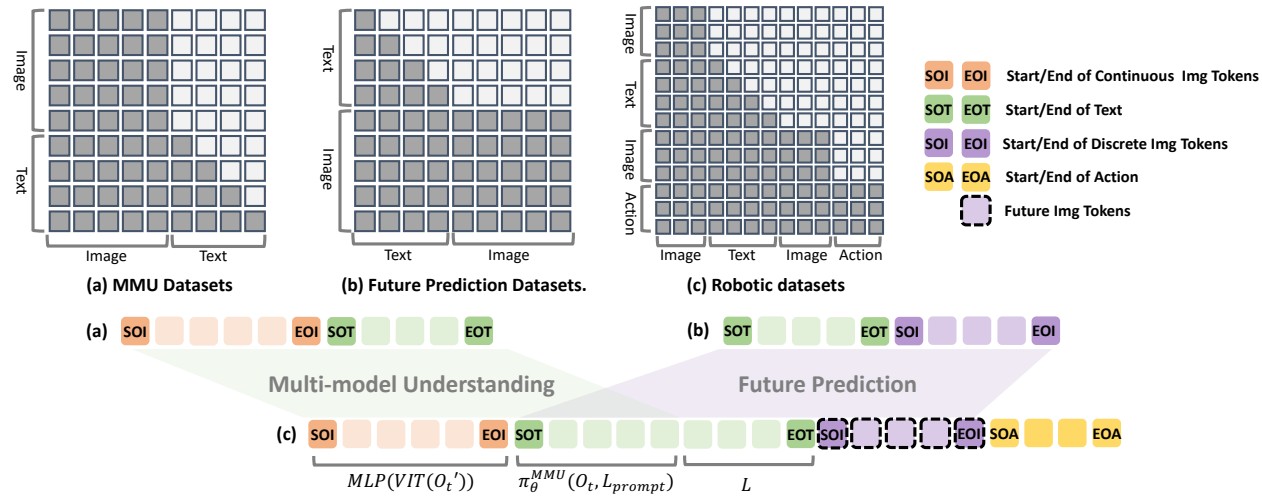

**Figure 4.** Illustration of the unified prompting and attention mechanism. We use special tokens to segment input sequences and identify task types. For MMU tasks, consecutive image tokens are placed before the language tokens, allowing image tokens to attend to one another. For image prediction, images are positioned after the language tokens, enabling them to attend to all prior information and predict future images that align with the language instructions. For action learning, which combines understanding and prediction, tokens from both tasks are concatenated, allowing actions to attend to both high-level scene understanding and low-level visual information.

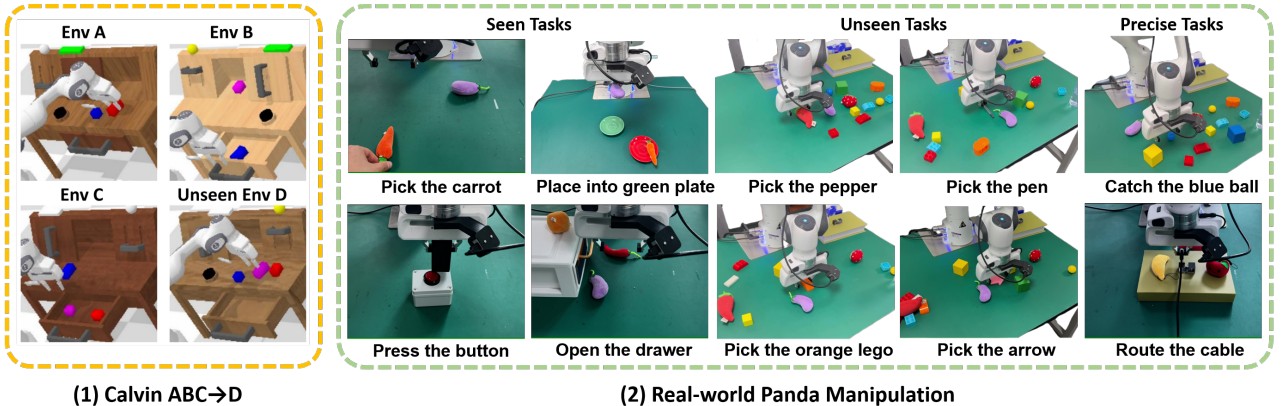

**Figure 5.** Visualization of our evaluation environments. The left is Calvin (Mees et al., 2022) in which we test on both ABC→D and ABCD→D settings. For real world, we train our model on simple tasks and test on more complex scenarios.

### 4.4.2. TRAINING OBJECTIVE

The UP-VLA method involves three modeling targets: language modeling for multi-modal understanding, image modeling for visual prediction, and action modeling for embodied tasks.

**Language Modeling for multi-modal Understanding.** Given $M$ visual tokens $\mathbf{u} = \{u_i\}_{i=0}^M$ and $N$ text tokens $\mathbf{l} = \{l_i\}_{i=0}^N$, we maximize the likelihood of the next token using cross-entropy loss:

$$\mathcal{L}_{MMU} = \sum_i \log p_\theta(l_i|\mathbf{u}, l_1, \cdots, l_{i-1})$$

**Image Modeling for Visual Prediction.** For the future image prediction task, given $M$ current image tokens $\mathbf{v}_t = \{v_i\}_{i=0}^M$ $N$ and instruction tokens $\mathbf{l} = \{l_i\}_{i=0}^N$, we use cross-entropy to predict future image tokens $\mathbf{v}_{t+\Delta t} = \{v'_i\}_{i=0}^M$:

$$\mathcal{L}_{PRE} = \sum_j \log p_\theta(v'_j|\mathbf{l}, v_1, \cdots, v_j, \cdots, v_M)$$

**Action Modeling for Embodied Tasks.** We minimize the mean squared error (MSE) between the predicted relative position $\hat{a}_{pos}$ and the ground-truth actions $a_{pos}$. The discrete status $a_{end}$ of the end-effector is optimized with binary cross-entropy loss (BCE):

$$\mathcal{L}_{ACT} = \sum ||\hat{a}_{pos} - a_{pos}||_2^2 + BCE(\hat{a}_{end}, a_{end})$$

We use varying weights to combine these three losses:

$$\mathcal{L} = \lambda_1 \mathcal{L}_{MMU} + \lambda_2 \mathcal{L}_{PRE} + \lambda_3 \mathcal{L}_{ACT}$$

# 5. Experiments

UP-VLA is a versatile vision-language-action model that can perform multi-modal understanding and future prediction while generating robot actions. In this section, we evaluate UP-VLA in two domains including the simulation CALVIN benchmark (Mees et al., 2022) and a real-world panda manipulation environment to verify the effectiveness of our UP-VLA framework.

## 5.1. Experiment Setup and baseline

**Setups.** For simulation evaluation, we utilize CALVIN (Mees et al., 2022), an open-source benchmark to learn long-horizon language-conditioned tasks. As shown in Figure 5(a), the CALVIN environment comprises 4 different scenes denoted ABCD. We evaluate UP-VLA on both ABCD-D and ABC-D settings.

Our real-world experiments involve multiple table-top manipulation tasks on the Franka-Emika Panda robot, including picking and placing, routing cables, pressing buttons, and opening drawers. Specifically, we collect over 2k demonstrations above 6 skills. As shown in Figure 5(b), we train UP-VLA on simple scenes while testing it on more complex settings. We place several seen and unseen objects on the table to introduce distractions and test whether the model can grasp entirely new objects to verify its semantic grounding capabilities. Meanwhile, we evaluate the model's ability to perform more fine-grained tasks, such as routing cables, grasping smaller unseen blocks, or picking up a pen. More details of dataset can be found in Appendix B.

**Baselines.** We mainly compare UP-VLA with two types of baselines: VLM-based VLA methods and future-prediction-based methods (mainly including future image generation, goal generation and video generation). All baselines in our experiment are listed as below:

- RT-1 (Brohan et al., 2022): a small robot action transformer using pretrained Efficient-Net (Tan & Le, 2019) as vision encoder.

- Diffusion Policy (Chi et al., 2023): a small action model using diffusion model.

- Robo-Flamingo (Li et al., 2023b): a typical VLA model consists of a pretrained VLM and an LSTM policy head.

- 3D-VLA (Zhen et al., 2024): a unified VLA model that enable 3D reasoning, multi-modal goal generation, and action planning. Different from UP-VLA, 3D-VLA mainly focuses on introducing 3D knowledge into VLM (LLM).

- UP-VLA-RT-2: an apple-to-apple baseline that makes use of the same backbone with UP-VLA and direct output actions autoregressively (reimplementation of

RT-2(Brohan et al., 2023) or OpenVLA (Kim et al., 2024)).

- Uni-Pi (Du et al., 2024): learns to generate future sequences and then output actions with an inverse kinematics model.

- Susie (Black et al., 2023): first generates goal image and then learns a goal-conditioned diffusion policy.

- GR-1 (Wu et al., 2023): pretrains a transformer on video prediction task and then finetune on robot data to learn multi-task robot manipulation.

- UP-VLA-phi-w/o-mmu: UP-VLA initialized from Phi-1.5 (Li et al., 2023c) and is trained without multi-modal understanding tasks, serves as a reimplemented GR-1 under the same setting of our method.

- 3D Diffuser Actor (Ke et al., 2024): learn a 3D diffusion policy using depth image with camera pose.

## 5.2. Simulation Evaluation

Table 1 and Table 2 presents the experimental results in simulation environments. UP-VLA achieves the highest performance on both ABC→D and ABCD→D tasks. Compared to other baselines, which perform significantly worse on ABC→D than on ABCD→D, UP-VLA achieves higher completion lengths in both scenarios, indicating that our method has better multitask learning and generalization capabilities in simulation tasks.

**Effectiveness of Visual Prediction.** Comparing VLA-based methods and prediction-based methods in Table 1, it is evident that previous VLA approaches underperform in simulation tasks relative to prediction-based methods. For example, RoboFlamingo, achieves a length of 2.47, which is less than GR-1's length of 3.06. This suggests that relying solely on vision-language understanding pretraining can be limiting in tasks that emphasize visual generalization. Our method addresses this limitation by incorporating visual prediction into the original VLA framework. Compared to UP-VLA-RT-2, which uses only action learning and achieves a completion length of 1.44, UP-VLA with visual prediction significantly improves the length to 4.08. This demonstrates that integrating visual prediction can substantially enhance the performance of original VLA methods.

**Effectiveness of multi-modal Understanding.** Compared to prediction-based methods, the UP-VLA method demonstrates superior performance. To further investigate the factors contributing to the performance improvement, we design UP-VLA-phi-w/o-mmu as a baseline to eliminate the variable of model backbone differences. This method initializes UP-VLA using a pure LLM, phi1.5 (Li et al., 2023c) and performs pretraining on the Bridge dataset for future prediction and is then finetuned with downstream robot task. Unlike UP-VLA, UP-VLA-phi-w/o-mmu does not include multi-modal understanding training, nor does it incorporate

| Method | Type | Task | Tasks completed in a row | | | | | |
|--------|------|------|---|---|---|---|---|---|
| | | | 1 | 2 | 3 | 4 | 5 | Avg. Len ↑ |
| RT-1 | other | ABC→D | 0.533 | 0.222 | 0.094 | 0.038 | 0.013 | 0.90 |
| Diffusion Policy* | other | ABC→D | 0.402 | 0.123 | 0.026 | 0.008 | 0.000 | 0.56 |
| 3D Diffuser Actor | other | ABC→D | **0.938** | 0.803 | 0.662 | 0.533 | 0.412 | 3.35 |
| 3D-VLA | VLA | ABC→D | 0.447 | 0.163 | 0.081 | 0.016 | 0.000 | 0.71 |
| UP-VLA-RT-2* | VLA | ABC→D | 0.612 | 0.389 | 0.236 | 0.138 | 0.062 | 1.44 |
| Robo-Flamingo | VLA | ABC→D | 0.824 | 0.619 | 0.466 | 0.331 | 0.235 | 2.47 |
| Uni-Pi | Prediction | ABC→D | 0.560 | 0.160 | 0.080 | 0.080 | 0.040 | 0.92 |
| Susie | Prediction | ABC→D | 0.870 | 0.690 | 0.490 | 0.380 | 0.260 | 2.69 |
| GR-1 | Prediction | ABC→D | 0.854 | 0.712 | 0.596 | 0.497 | 0.401 | 3.06 |
| UP-VLA-phi-w/o-mmu* | Prediction | ABC→D | 0.844 | 0.705 | 0.604 | 0.520 | 0.430 | 3.13 |
| **UP-VLA** | Prediction&VLA | ABC→D | 0.928 | **0.865** | **0.815** | **0.769** | **0.699** | **4.08** |

*Table 1.* Zero-shot long-horizon evaluation on the Calvin benchmark where agent is asked to complete five chained tasks sequentially. Results marked with an asterisk (*) indicate those that we reproduced, while the rest are copied from the original papers. UP-VLA achieves the best performance, demonstrating that our approach exhibits strong generalization capabilities in simulated environments.

| Method | Type | Task | Tasks completed in a row | | | | | |
|--------|------|------|---|---|---|---|---|---|
| | | | 1 | 2 | 3 | 4 | 5 | Avg. Len ↑ |
| RT-1 | other | ABCD→D | 0.844 | 0.617 | 0.438 | 0.323 | 0.227 | 2.45 |
| Robo-Flamingo | VLA | ABCD→D | **0.964** | 0.896 | 0.824 | 0.740 | 0.660 | 4.09 |
| GR-1 | Prediction | ABCD→D | 0.949 | 0.896 | 0.844 | 0.789 | 0.731 | 4.21 |
| **UP-VLA** | Prediction&VLA | ABCD→D | 0.962 | **0.921** | **0.879** | **0.842** | **0.812** | **4.42** |

*Table 2.* Zero-shot long-horizon evaluation on the Calvin ABCD→D benchmark. Results of baselines are copied from original papers.

image comprehension in the language prompts during output. As shown in Table 1, UP-VLA-phi-w/o-mmu performs worse than UP-VLA, which indicates that injecting multi-modal understanding into the model during training helps improve its generalization ability in new scenarios.

### 5.3. Real Robot Evaluation

For real-world experimental results, we train RT-1 (Brohan et al., 2022), Diffusion Policy (Chi et al., 2023) on our datasets (using the open-source code and testing them on the same physical hardware). To ensure a fair comparison, we reproduce RT-2 (Brohan et al., 2023) and GR-1 (Wu et al., 2023) using our dataset and backbone, referred to as UP-VLA-RT-2 and UP-VLA-phi-w/o-mmu, respectively. We report the success rate of each task over 20 attempts during real-world roll-out.

Figure 6 presents the results of our evaluation on three types of real-world tasks. UP-VLA demonstrates significant improvement across all tasks. Specifically, for tasks seen during training, as shown on the left side of Figure 5(b), the three methods based on the UP-VLA backbone outperform RT-1 and Diffusionpolicy, indicating that using LLM backbone exhibit superior multitasking capabilities.

For unseen tasks, we first test the ability to grasp new objects

not encountered during training, as shown in the middle of Figure 5. UP-VLA-RT-2 outperforms UP-VLA-phi-w/o-mmu, suggesting that multi-modal understanding aids semantic generalization ability. UP-VLA demonstrates better visual-semantic generalization for these tasks, proving that our approach effectively aligns multi-modal understanding with objects and actions. The right side of Figure 6 shows the performance on tasks requiring more precise operations (e.g., routing cables, grasping small objects, and picking up previously unseen pens). These tasks demand the model's enhanced spatial understanding and ability to perceive visual details. As opposed to new objects, UP-VLA-Phi-w/o-mmu excels at precise operations compared to UP-VLA-RT-2. UP-VLA performs best on these tasks, highlighting that the integration of future visual prediction enhances VLA's understanding of physical space and details.

### 5.4. Ablation Studies

In this section, we aim to understand each module in UP-VLA. We compare the full UP-VLA with the following methods: UP-VLA-w/o-MMU, which does not utilize the LLava tuning dataset for multi-modal understanding, UP-VLA-w/o-Bridge-Pretrain, which skips visual prediction pretraining on the bridge dataset; UP-VLA-w/o-Prediction, which bypasses visual prediction and directly learns actions;

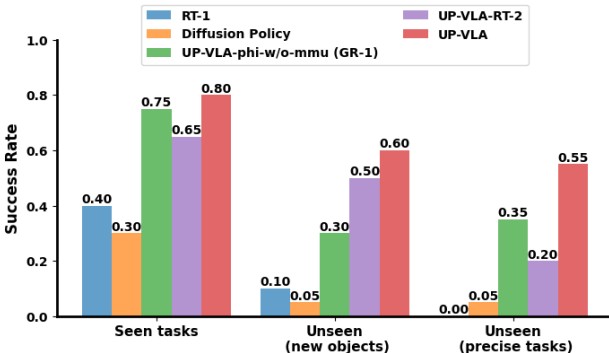

*Figure 6.* Results on real-world manipulation tasks.

| Method | ABC→D | Real World | |
|---|---|---|---|
| | Avg.Len↑ | Seen↑ | Unseen↑ |
| w/o MMU | 3.89 | 0.85 | 0.20 |
| w/o Bridge-Pretrain | 2.74 | 0.65 | 0.30 |
| w/o Prediction | 1.44 | 0.65 | 0.35 |
| w/o MMU-Condition | 3.99 | 0.80 | 0.50 |
| Full | 4.08 | 0.80 | 0.58 |

*Table 3.* Ablating components of UP-VLA.

and UP-VLA-w/o-MMU-Condition, which omits the mechanism described in sec 4.3 that extends visual prediction prompts using MMU. Table 3 presents the performance of different ablation methods on simulated Calvin tasks and real-world robot tasks.

**How does the visual prediction mechanism affect the performance of UP-VLA?** After removing the visual prediction task, performance of UP-VLA in Calvin drops from 4.08 to 1.44. This indicates that stronger visual supervision contributes to the model's ability to generalize in new environments. Furthermore, omitting the visual prediction pretraining on the bridge dataset also led to noticeable performance degradation, highlighting the importance of the pretraining step for the VLM to learn the dynamics of the physical world effectively.

**Does multi-modal understanding enhance the generalization ability of the model?** In real-world experiments, removing MMU tasks or MMU-condition mechanisms from training leads to comparable or higher performance on seen tasks but results in a decline in performance on unseen objects. This observation indicates that joint training with MMU and using MMU to augment input prompts are crucial for semantic generalization and help prevent the model from overfitting to the dataset.

### 5.5. Quantitative Results

Figure 7 visualizes the performance of UP-VLA in multimodal understanding question-answering (VQA) and future prediction across different types of data.

**Multi-modal Understanding** As shown in Fig 7, it can be observed that UP-VLA can identify the objects present in embodied scenes and estimate their approximate relative positions, which is critical for action learning. Therefore, effectively integrating MMU capabilities with action learning is a promising approach to improving operational accuracy. Additionally, we observe that the model's identification of the specific objects is sometimes inaccurate. This is likely due to constraints in the scale of data and backbone, which is a potential direction for future research.

**Future Prediction** Results of predicted images are shown in Fig 7. UP-VLA demonstrates the ability to accurately predict the positions of robot arms and objects based on language instructions. However, we also identify some limitations. For instance, in the Calvin D environment, the predicted frames feature background colors that differ from the input frames. The model tends to use background colors from the training datasets (ABC). This issue is likely due to insufficient pretraining for visual generation, which limits the model's generalization capability in generation tasks.

## 6. Conclusion

In this paper, we introduce UP-VLA, a vision-language-action model that can understand, generate predicted future images, and plan actions in the embodied environment. We devise a novel VLA training paradigm that unifies policy learning with visual prediction and multi-modal understanding. Our results show that the use of future image prediction can significantly improve the precision and visual generalization of policy. We also further enhance our model by introducing multi-modal understanding knowledge to visual prediction-based policy learning, which demonstrates stronger generalization ability in both semantic grounding and spatial understanding.

## Impact Statement

This paper presents a novel research to advance robot manipulation models with unifed training strategies. Given that robots operate in the physical world under certain human instructions, high-level semantic content and also low-level visual and spatial details are crucial for accurate robot control. To mitigate this issue, our approach involves unified training paradigm to force VLA to capture both semantic information and learn dynamics of the physical world.

## References

Balazadeh, V., Ataei, M., Cheong, H., Khasahmadi, A. H., and Krishnan, R. G. Synthetic vision: Training vision-language models to understand physics. *arXiv preprint arXiv:2412.08619*, 2024.

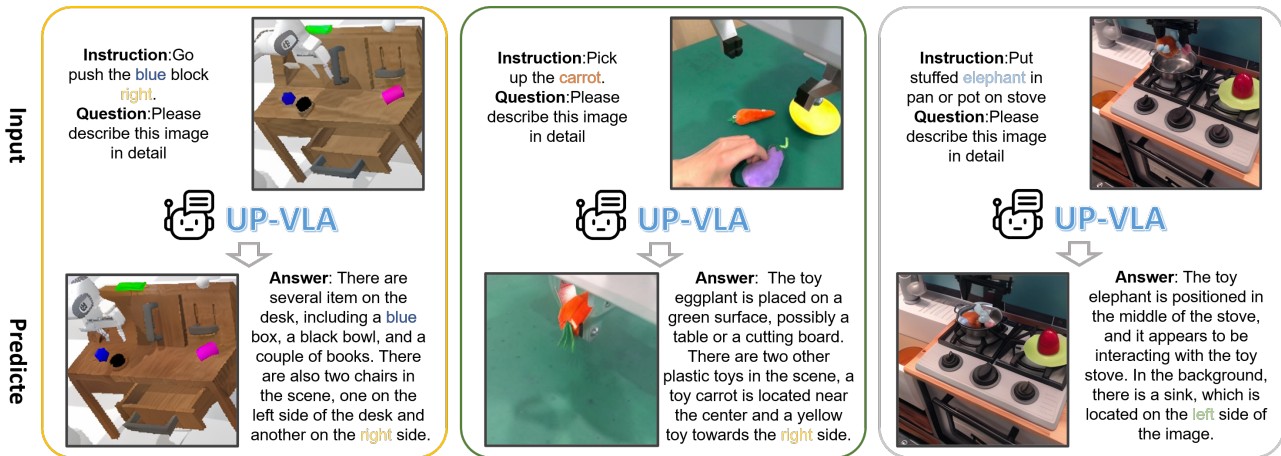

*Figure 7.* Visualization of VQA results and predicted future images.

Black, K., Nakamoto, M., Atreya, P., Walke, H., Finn, C., Kumar, A., and Levine, S. Zero-shot robotic manipulation with pretrained image-editing diffusion models. *arXiv preprint arXiv:2310.10639*, 2023.

Black, K., Brown, N., Driess, D., Esmail, A., Equi, M., Finn, C., Fusai, N., Groom, L., Hausman, K., Ichter, B., et al. $pi\_0$: A vision-language-action flow model for general robot control. *arXiv preprint arXiv:2410.24164*, 2024.

Blattmann, A., Dockhorn, T., Kulal, S., Mendelevitch, D., Kilian, M., Lorenz, D., Levi, Y., English, Z., Voleti, V., Letts, A., et al. Stable video diffusion: Scaling latent video diffusion models to large datasets. *arXiv preprint arXiv:2311.15127*, 2023.

Brohan, A., Brown, N., Carbajal, J., Chebotar, Y., Dabis, J., Finn, C., Gopalakrishnan, K., Hausman, K., Herzog, A., Hsu, J., et al. Rt-1: Robotics transformer for real-world control at scale. *arXiv preprint arXiv:2212.06817*, 2022.

Brohan, A., Brown, N., Carbajal, J., Chebotar, Y., Chen, X., Choromanski, K., Ding, T., Driess, D., Dubey, A., Finn, C., et al. Rt-2: Vision-language-action models transfer web knowledge to robotic control. *arXiv preprint arXiv:2307.15818*, 2023.

Chen, B., Xu, Z., Kirmani, S., Ichter, B., Sadigh, D., Guibas, L., and Xia, F. Spatialvlm: Endowing vision-language models with spatial reasoning capabilities. In *Proceedings of the IEEE/CVF Conference on Computer Vision and Pattern Recognition*, pp. 14455–14465, 2024a.

Chen, X., Guo, J., He, T., Zhang, C., Zhang, P., Yang, D. C., Zhao, L., and Bian, J. Igor: Image-goal representations are the atomic control units for foundation models in embodied ai, 2024b. URL https://arxiv.org/abs/2411.00785.

Chi, C., Feng, S., Du, Y., Xu, Z., Cousineau, E., Burchfiel, B., and Song, S. Diffusion policy: Visuomotor policy learning via action diffusion. In *Proceedings of Robotics: Science and Systems (RSS)*, 2023.

Cui, C., Ding, P., Song, W., Bai, S., Tong, X., Ge, Z., Suo, R., Zhou, W., Liu, Y., Jia, B., et al. Openhelix: A short survey, empirical analysis, and open-source dual-system vla model for robotic manipulation. *arXiv preprint arXiv:2505.03912*, 2025.

Dai, W., Li, J., Li, D., Tiong, A. M. H., Zhao, J., Wang, W., Li, B., Fung, P. N., and Hoi, S. Instructblip: Towards general-purpose vision-language models with instruction tuning. *Advances in Neural Information Processing Systems*, 36, 2024.

Ding, P., Zhao, H., Zhang, W., Song, W., Zhang, M., Huang, S., Yang, N., and Wang, D. Quar-vla: Vision-language-action model for quadruped robots. In *European Conference on Computer Vision*, pp. 352–367. Springer, 2024.

Ding, P., Ma, J., Tong, X., Zou, B., Luo, X., Fan, Y., Wang, T., Lu, H., Mo, P., Liu, J., et al. Humanoid-vla: Towards universal humanoid control with visual integration. *arXiv preprint arXiv:2502.14795*, 2025.

Dosovitskiy, A., Beyer, L., Kolesnikov, A., Weissenborn, D., Zhai, X., Unterthiner, T., Dehghani, M., Minderer, M., Heigold, G., Gelly, S., et al. An image is worth 16x16 words: Transformers for image recognition at scale. *arXiv preprint arXiv:2010.11929*, 2020.

Driess, D., Xia, F., Sajjadi, M. S., Lynch, C., Chowdhery, A., Ichter, B., Wahid, A., Tompson, J., Vuong, Q., Yu, T., et al. Palm-e: An embodied multimodal language model. *arXiv preprint arXiv:2303.03378*, 2023.

Du, Y., Yang, S., Dai, B., Dai, H., Nachum, O., Tenenbaum, J., Schuurmans, D., and Abbeel, P. Learning universal

policies via text-guided video generation. *Advances in Neural Information Processing Systems*, 36, 2024.

Esser, P., Rombach, R., and Ommer, B. Taming transformers for high-resolution image synthesis. In *Proceedings of the IEEE/CVF conference on computer vision and pattern recognition*, pp. 12873–12883, 2021.

Ge, Y., Zhao, S., Zhu, J., Ge, Y., Yi, K., Song, L., Li, C., Ding, X., and Shan, Y. Seed-x: Multimodal models with unified multi-granularity comprehension and generation. *arXiv preprint arXiv:2404.14396*, 2024.

Ghaffari, S. and Krishnaswamy, N. Exploring failure cases in multimodal reasoning about physical dynamics. In *Proceedings of the AAAI Symposium Series*, volume 3, pp. 105–114, 2024.

Guo, Y., Hu, Y., Zhang, J., Wang, Y.-J., Chen, X., Lu, C., and Chen, J. Prediction with action: Visual policy learning via joint denoising process. *arXiv preprint arXiv:2411.18179*, 2024.

Ho, J., Jain, A., and Abbeel, P. Denoising diffusion probabilistic models. *Advances in neural information processing systems*, 33:6840–6851, 2020.

Jang, E., Irpan, A., Khansari, M., Kappler, D., Ebert, F., Lynch, C., Levine, S., and Finn, C. Bc-z: Zero-shot task generalization with robotic imitation learning. In *Conference on Robot Learning*, pp. 991–1002. PMLR, 2022.

Ke, T.-W., Gkanatsios, N., and Fragkiadaki, K. 3d diffuser actor: Policy diffusion with 3d scene representations. *arXiv preprint arXiv:2402.10885*, 2024.

Kim, M., Pertsch, K., Karamcheti, S., Xiao, T., Balakrishna, A., Nair, S., Rafailov, R., Foster, E., Lam, G., Sanketi, P., Vuong, Q., Kollar, T., Burchfiel, B., Tedrake, R., Sadigh, D., Levine, S., Liang, P., and Finn, C. Openvla: An open-source vision-language-action model. *arXiv preprint arXiv:2406.09246*, 2024.

Li, J., Li, D., Savarese, S., and Hoi, S. Blip-2: Bootstrapping language-image pre-training with frozen image encoders and large language models. In *International conference on machine learning*, pp. 19730–19742. PMLR, 2023a.

Li, X., Liu, M., Zhang, H., Yu, C., Xu, J., Wu, H., Cheang, C., Jing, Y., Zhang, W., Liu, H., et al. Vision-language foundation models as effective robot imitators. *arXiv preprint arXiv:2311.01378*, 2023b.

Li, Y., Bubeck, S., Eldan, R., Del Giorno, A., Gunasekar, S., and Lee, Y. T. Textbooks are all you need ii: phi-1.5 technical report. *arXiv preprint arXiv:2309.05463*, 2023c.

Li, Z., Wang, H., Liu, D., Zhang, C., Ma, A., Long, J., and Cai, W. Multimodal causal reasoning benchmark: Challenging vision large language models to infer causal links between siamese images. *arXiv preprint arXiv:2408.08105*, 2024.

Liu, H., Li, C., Wu, Q., and Lee, Y. J. Visual instruction tuning. *Advances in neural information processing systems*, 36, 2024.

Mees, O., Hermann, L., Rosete-Beas, E., and Burgard, W. Calvin: A benchmark for language-conditioned policy learning for long-horizon robot manipulation tasks. *IEEE Robotics and Automation Letters (RA-L)*, 7(3):7327–7334, 2022.

O'Neill, A., Rehman, A., Gupta, A., Maddukuri, A., Gupta, A., Padalkar, A., Lee, A., Pooley, A., Gupta, A., Mandlekar, A., et al. Open x-embodiment: Robotic learning datasets and rt-x models. *arXiv preprint arXiv:2310.08864*, 2023.

Radford, A., Kim, J. W., Hallacy, C., Ramesh, A., Goh, G., Agarwal, S., Sastry, G., Askell, A., Mishkin, P., Clark, J., et al. Learning transferable visual models from natural language supervision. In *International conference on machine learning*, pp. 8748–8763. PMLR, 2021.

Song, W., Chen, J., Ding, P., Zhao, H., Zhao, W., Zhong, Z., Ge, Z., Ma, J., and Li, H. Accelerating vision-language-action model integrated with action chunking via parallel decoding. *arXiv preprint arXiv:2503.02310*, 2025.

Tan, M. and Le, Q. Efficientnet: Rethinking model scaling for convolutional neural networks. In *International conference on machine learning*, pp. 6105–6114. PMLR, 2019.

Walke, H., Black, K., Lee, A., Kim, M. J., Du, M., Zheng, C., Zhao, T., Hansen-Estruch, P., Vuong, Q., He, A., Myers, V., Fang, K., Finn, C., and Levine, S. Bridgedata v2: A dataset for robot learning at scale. In *Conference on Robot Learning (CoRL)*, 2023.

Wang, W., Bao, H., Dong, L., Bjorck, J., Peng, Z., Liu, Q., Aggarwal, K., Mohammed, O. K., Singhal, S., Som, S., et al. Image as a foreign language: Beit pretraining for all vision and vision-language tasks. *arXiv preprint arXiv:2208.10442*, 2022.

Wang, Y.-J., Zhang, B., Chen, J., and Sreenath, K. Prompt a robot to walk with large language models. *arXiv preprint arXiv:2309.09969*, 2023.

Wen, C., Jayaraman, D., and Gao, Y. Can transformers capture spatial relations between objects? *arXiv preprint arXiv:2403.00729*, 2024.

Wu, H., Jing, Y., Cheang, C., Chen, G., Xu, J., Li, X., Liu, M., Li, H., and Kong, T. Unleashing large-scale video generative pre-training for visual robot manipulation. *arXiv preprint arXiv:2312.13139*, 2023.

Xie, J., Mao, W., Bai, Z., Zhang, D. J., Wang, W., Lin, K. Q., Gu, Y., Chen, Z., Yang, Z., and Shou, M. Z. Show-o: One single transformer to unify multimodal understanding and generation. *arXiv preprint arXiv:2408.12528*, 2024.

Yu, L., Cheng, Y., Sohn, K., Lezama, J., Zhang, H., Chang, H., Hauptmann, A. G., Yang, M.-H., Hao, Y., Essa, I., et al. Magvit: Masked generative video transformer. In *Proceedings of the IEEE/CVF Conference on Computer Vision and Pattern Recognition*, pp. 10459–10469, 2023.

Zhang, J., Guo, Y., Chen, X., Wang, Y.-J., Hu, Y., Shi, C., and Chen, J. Hirt: Enhancing robotic control with hierarchical robot transformers. *arXiv preprint arXiv:2410.05273*, 2024.

Zhao, W., Ding, P., Zhang, M., Gong, Z., Bai, S., Zhao, H., and Wang, D. Vlas: Vision-language-action model with speech instructions for customized robot manipulation. *arXiv preprint arXiv:2502.13508*, 2025.

Zhen, H., Qiu, X., Chen, P., Yang, J., Yan, X., Du, Y., Hong, Y., and Gan, C. 3d-vla: A 3d vision-language-action generative world model. *arXiv preprint arXiv:2403.09631*, 2024.

Zheng, B., Gu, J., Li, S., and Dong, C. Lm4lv: A frozen large language model for low-level vision tasks. *arXiv preprint arXiv:2405.15734*, 2024a.

Zheng, R., Liang, Y., Huang, S., Gao, J., Daumé III, H., Kolobov, A., Huang, F., and Yang, J. Tracevla: Visual trace prompting enhances spatial-temporal awareness for generalist robotic policies. *arXiv preprint arXiv:2412.10345*, 2024b.

# A. Implementation Details

We use pretrained Showo-512x512(1.3B) (Xie et al., 2024) as the backbone and CLIP-VIT (Radford et al., 2021), MagVIT (Yu et al., 2023) (VQ-GAN (Esser et al., 2021)). In the pretrain stage, we train UP-VLA for 20k steps with batch size of 64 on future prediction and vision-language understanding tasks. We apply a linear warmup at the first 1k steps. In the action learning stage, we train UP-VLA with a batch size of 64.

# B. Manipulation Dataset Details

For the simulation environment data, we follow the setups of the CALVIN benchmark (using its training sets and evaluation sets).

In the real world, our training data is collected using both manual and scripted methods. Specifically, for tasks such as grasping two types of toy fruits (carrot and eggplant), opening drawers, and routing cable tasks, we collect demonstrations manually using a remote operation joystick, ensuring that the target objects are roughly evenly distributed in the field of view. For grasping blocks of different colors, we use scripted policies. We randomly initialize the robotic arm's position to collect trajectories for these tasks.

For real-world evaluation, our tests primarily focus on unseen settings to validate the semantic generalization capability of our method. We test whether the model can complete tasks despite the introduction of more distracting objects, a broader range of positional variations, diverse backgrounds, and entirely new objects, e.g. different shapes of vegetables, an arrow-shaped paper, unseen vegetables, toy pizza, and blocks with unseen color. We also design precise tasks to better test model's ability to handle visual details, like picking up a small ball or pen.

