# OpenReview forum: "UP-VLA:  A Unified Understanding and Prediction Model for Embodied Agent"
_ICML.cc/2025/Conference — ICML 2025 poster_

### Official Review · Reviewer_FLHB · 2025-03-11

**Overall Recommendation:** 4

**Summary:**

This paper presents a new training paradigm for Vision-Language-Action (VLA) models by training with both multi-modal understanding and future prediction objectives. Multi-modal Understanding is in the form of question-answering given a paired image-text, and the future visual prediction aims to predict the target image given the current image and instruction pair. Prior research usually tackles one of these two, which the authors frame as "high-level semantic content" and "low-level feature". Experiments are conducted on the simulation CALVIN benchmark and a real-world manipulation environment to verify the effectiveness of our UP-VLA framework.

## post rebuttal
I appreciate the authors' rebuttal, especially about pretraining and extra experiment results. After reading opinions from other reviewers, I think the paper presents clear evidence that pretraining with mixed multi-modal understanding and future prediction objectives helps the VLA models, and will keep my original recommendation as accept.

**Claims And Evidence:**

I think the major claim of the paper "combining both vision-language understanding and future prediction objectives, enabling the capture of both high-level semantic and low-level visual patterns essential for embodied agent" is verified by the experimental results compared with the baselines on the CALVIN benchmark and real-world manipulation tasks. It is further verified by the ablative study, where a significant difference is observed when dropping either of them.

**Essential References Not Discussed:**

Not that I'm aware of.

**Experimental Designs Or Analyses:**

The experiment results on the CALVIN benchmark demonstrate the proposed outperforms the listed baselines on the zero-shot evaluation. However, this cannot fully verify the proposed dual-objective pretraining as the pretrained data and model architecture vary across different methods. This is similar in real-world manipulation tasks. The results difference looks statistically significant in both settings. The ablation results more evidently demonstrate the effectiveness of the proposed pretraining paradigm.

**Methods And Evaluation Criteria:**

While visual prediction and language prediction are quite mature in model architecture on their own, and there are existing models that aim to make multi-modal predictions, I believe the proposed method, especially the unified prompting and attention mechanism, is reasonable for verifying the effectiveness of different pre-training objectives.

**Other Comments Or Suggestions:**

Missing punctuation (periods or commas) in the equations.

**Other Strengths And Weaknesses:**

[+] I find the motivation of this paper to introduce both multi-model and visual prediction as pretraining is valid, the architecture design is reasonable and the experiments verify its effectiveness.

[-]  I'm curious about the quantitative performance of multi-modal and visual prediction after the pretraining stage, on top of the action performance currently presented in the paper. Could the model address some multi-model tasks directly? How is the performance of next-frame prediction, working as a world model? I think providing such evaluations will further help people understand how the model performs before and after the fine-tuning stage.

[-] Another thing I find interesting, but currently missing in the paper is potentially the insights from the pretraining. Do the two objectives conflict with each other? Or is the pretraining simply just working?

**Questions For Authors:**

See the strengths and weaknesses section.

**Relation To Broader Scientific Literature:**

I think to improve the effectiveness and generalizability is a major focus of the VLA models in the era of embodied AI, and I think the major problem the paper aims to tackle is relevant and important.

**Theoretical Claims:**

No major theoretical claims are made in the paper.

---

> ### Author Rebuttal · Authors · 2025-04-01
>
> We sincerely appreciate your time and efforts in reviewing our paper! Based on your review, we added a detailed discussion.
>
> ---
>
> **Q1: I'm curious about the quantitative performance of multi-modal and visual prediction after the pretraining stage, on top of the action performance currently presented in the paper.**
>
> ANS: We would like to clarify that our primary focus is on robot decision-making. Due to limited computational resources, our pretraining stage uses the LLaVA instruction-tuning dataset for multi-modal understanding (MMU) training and the Bridge robot dataset for future visual prediction training. Our pretraining process adapts the general-purpose Show-o model to robotic domain-specific tasks, which may come at the cost of reduced performance in general settings, such as VQA, OCR, and text-to-image generation. In our approach, we expect the VLM to learn visual predictions that are consistent with language. Although this approach may lead to the loss of ability on general tasks, it will be more conducive to visual prediction and low-level action learning. Results on both simulation and real-world show that this adaptation significantly benefits robot control tasks.
>
>
>
> ---
>
> **Q2: Another thing I find interesting, but currently missing in the paper is potentially the insights from the pretraining. Do the two objectives conflict with each other? Or is the pretraining simply just working?**
>
>
> ANS: Thank you for your insightful comments! Pretraining models for both multi-modal understanding and generation is becoming increasingly popular. Many works in the generative modeling field have explored mixed-pretraining approaches, such as TransFusion [1] and Show-o [2]. Their original objective functions focus on multi-modal understanding and text-to-image generation.
>
> Inspired by the success of these models, we designed an objective function tailored for embodied agents, combining MMU tasks with future image prediction. Since our model builds upon the Show-o framework, we observed that pretraining proceeds without conflicts, and the training remains stable without collapse.
>
> [1] Transfusion: Predict the Next Token and Diffuse Images with One Multi-Modal Model https://arxiv.org/abs/2408.11039
>
> [2]Show-o: One Single Transformer to Unify Multimodal Understanding and Generation https://arxiv.org/abs/2408.12528
>
> ---
>
> Thank you again for your time and effort in reviewing our work! We hope this clarification can solve your concerns!

---

### Official Review · Reviewer_dQTi · 2025-03-12

**Overall Recommendation:** 3

**Summary:**

The paper presents UP-VLA, a unified Vision-Language-Action (VLA) model designed for embodied agents. The model aims to enhance both high-level semantic comprehension and low-level spatial understanding by integrating multi-modal understanding and future prediction objectives while current VLMs focus on high-level semantics while neglecting fine-grained spatial and dynamic features.

**Claims And Evidence:**

yes

**Essential References Not Discussed:**

This paper lacks a comparison with the latest VLA models.

**Experimental Designs Or Analyses:**

Please refer to 'Other Strengths and Weaknesses.'

**Methods And Evaluation Criteria:**

Please refer to 'Other Strengths and Weaknesses.'

**Other Comments Or Suggestions:**

Please refer to 'Other Strengths and Weaknesses.'

**Other Strengths And Weaknesses:**

Strengths:

1. The paper proposes a unified training approach that merges semantic understanding and future prediction, incorporating three types of datasets —— Vision-Language datasets for high-level semantic understanding, Internet video datasets for low-level visual dynamics, and Robotic action datasets for embodied control. Through the constructed dataset and training strategy, UP-VLA explicitly enhances physical spatial comprehension.

2. UP-VLA achieved promising results in the Calvin simulation benchmark, demonstrating strong long-horizon manipulation capabilities.



Weaknesses:

1. The primary concern is the model's efficiency. While incorporating future image prediction before action prediction can significantly enhance performance, it raises the question of whether this autoregressive next-token prediction paradigm could lead to a substantial decrease in inference speed. The author's exploration is interesting, but inference speed plays a critical role in determining the robot's control frequency, which is essential in robotics.

2. When pretraining on an internet video dataset, if robotic videos are used for future prediction, why not pretrain action prediction simultaneously? For example, leveraging large-scale simulator data or real-world robotic data (e.g., Open X-Embodied or DROID).

3. The paper's writing is incomplete and lacks many details. For example: Which internet videos were used for pretraining? Why are human hands present in Figure 5? Why is there no execution video demonstration?

4. Lacks comparison with SOTA autoregressive-based VLA methods, such as OpenVLA.

**Questions For Authors:**

Please refer to 'Other Strengths and Weaknesses.'

**Relation To Broader Scientific Literature:**

Effectively constructing a VLA model for robotics.

**Theoretical Claims:**

Application paper without making theoretical claims.

---

> ### Author Rebuttal · Authors · 2025-04-01
>
> We sincerely appreciate your time and efforts in reviewing our paper! Based on your review, we added a detailed discussion and additional experiments.
>
> ---
>
> **Q1: About model efficiency**
>
> ANS:
> We would like to clarify that UP-VLA operates at **almost the same control frequency as previous VLA methods**. We did not use autoregressive image generation, we **jointly predict future images and actions** with a single forward LLM pass during policy execution, resulting in almost similar computational cost to prior VLA models. The table below presents runtime efficiency based on the same backbone (Showo), comparing models with and without image prediction (measured as the reciprocal of the average inference time over 100 iterations on the same hardware):
>
> | Method        | Inference Speed ↑ |
> |---------------|-------------------|
> | UP-VLA-RT-2*  | 13.2Hz            |
> | UP-VLA        | 13.0Hz            |
>
> Additionally, our method employs an action chunking size of 10. We observed that action chunk can further enhances performance and produces smoother trajectories.
>
>
> ---
>
> **Q2: When pretraining on an internet video dataset, if robotic videos are used for future prediction, why not pretrain action prediction simultaneously?**
>
>
> ANS: Thank you for your question. We did not include action prediction in pretraining stage based on these two reasons: (1) Different robot datasets have varying action spaces with distinct meanings, requiring additional design effort to standardize them under a unified output format. (2) We believe that robot trajectory videos provide more consistent representations across datasets and already contain valuable dynamics information.
>
> Following your suggestion, we conducted an additional experiment incorporating robot action learning during pretraining while keeping the architecture unchanged (by padding actions to the same dimensions). The results indicate that this modification provides no benefit. We hypothesize that this is because RGB images already contain sufficient prior information, and the differences in action spaces across robots make direct action prediction less effective.
>
> | CALVIN ABC-D        | Avg Len ↑ |
> |---------------|-------------------|
> | UP-VLA              | 4.08      |
> | UP-VLA with Action Pretraining        |  3.98            |
>
>
> ---
>
> **Q3: About details: Which internet videos were used for pretraining? Why are human hands present in Figure 5? Why is there no execution video demonstration?**
>
> ANS: We use the Bridge dataset for image-prediction pretraining, as mentioned in Section 5.2. The human hands in Figure 5 serve as disturbances during policy rollout. For your convenience, we have included additional execution details in the following anonymous video links: https://sites.google.com/view/upvla-rebuttal.
>
> ---
> **Q4: About SOTA autoregressive-based VLA methods**
>
> ANS: Following your suggestions, we conducted experiments on the influential models pi0, openvla, and octo. Since these methods do not have official test results, we utilize the open-source code to fine-tune and test them on the calvin data (with settings consistent with UP-VLA). The results on the ABC-D task are shown in the table below, where an asterisk indicates the results we reproduced using the open-source code.
>
> |  Method  |      Type      | Avg. Len ↑ |
> | :------: | :------------: | :--------: |
> | Openvla* |      VLA       |    1.60    |
> |  Octo*   |      VLA       |    0.59    |
> |   pi0*   |      VLA       |    3.63    |
> |  UP-VLA  | Prediction&VLA |    4.08    |
>
> ---
>
> Thank you again for your time and effort in reviewing our work! We hope our clarification can solve all your concerns and shows the improved quality of our paper!

---

> > ### Comment · Reviewer_dQTi · 2025-04-08
> >
> > Thank you for the rebuttal. I still have two questions I would like to discuss with the authors:
> >
> > 1. Is the inference speed provided by the authors the same as the model's inference speed, or is it the model's inference speed multiplied by the action chunk size?
> >
> > 2. Can a detailed manipulation success rate be provided for comparison with other SOTA autoregressive-based VLA methods?
> >
> >
> > -------
> > Thank you for the authors' response (as shown in below). The responses address all of my previous concerns, and I will raise my rating to "weakly accept." Finally, I hope the authors can perform further real-world validation on a broader range of long-horizon and non-pick-and-place tasks.

---

> > > ### Author Response · Authors · 2025-04-08
> > >
> > > Dear Reviewer dQTi:
> > >
> > > We are delighted to receive your response! We are very gald to provide further details in the blow:
> > >
> > > ---
> > >
> > >
> > > Q1: Is the inference speed provided by the authors the same as the model's inference speed, or is it the model's inference speed multiplied by the action chunk size?
> > >
> > > ANS: **The inference speed is the same as the model’s forward inference speed.** A forward path of a 1.3B VLM model cost about ~80ms which result in 13Hz control frequency. If we multiply the action chunk size (which is set to 10 in our configuration), the final frequency will be approximately 130 Hz. More detailed information are provided below:
> > >
> > > |                    | Inference Speed (without action chunk) | Action Frequency (with action chunk)       |
> > > | ------------------ | -------------------------------------- | ------------------------------------------ |
> > > | Openvla (7B)       | ~4Hz                                   | ~80Hz (action chunk-25 in **OpenVLA-oft**) |
> > > | UP-VLA-RT-2 (1.5B) | ~13Hz                                  | /                                          |
> > > | UP-VLA (1.5B)      | ~13Hz                                  | ~130Hz  (action chunk-10)                  |
> > >
> > > Since we **jointly predict the image and actions within a single LLM forward pass**, the model’s inference speed is nearly the same as that of previous VLA methods.
> > >
> > >
> > > ---
> > >
> > > Q2: Can a detailed manipulation success rate be provided for comparison with other SOTA autoregressive-based VLA methods?
> > >
> > > Ans: Yes! We are happy to provide a more detailed success rate for comparison on the standard simulated CALVIN ABC-D benchmark.
> > >
> > > The CALVIN ABC-D benchmark consists of 34 different types of manipulation tasks, where agents are required to complete five random-selected tasks sequentially, following a given instruction.
> > > Here is the detailed manipulation success rate.
> > >
> > > |  **Success Rates** |      Type      |   1_st task   | 2_nd task     | 3_th task     | 4_th task     | 5_th tasks     | Num Task Success ↑ |
> > > | :------: | :------------: | :---: | ----- | ----- | ----- | ----- | ---------- |
> > > | Openvla* |      VLA       | 73.1\% | 42.4\% | 24.0\% | 12.9\% | 7.5\% | 1.60
> > > |  Octo*   |      VLA       | 46.6\% | 11.1\% | 1.6\% | 0.1\% | 0.0\% | 0.59       |
> > > |   pi0*   |      VLA       | 91.6\% | 82.1\% | 71.7\% | 64.1\% | 53.8\% | 3.63       |
> > > |  UP-VLA  | Prediction&VLA | 92.8 \%| 86.5\% | 81.5\% | 76.9\% | 69.9\% | 4.08       |
> > >
> > > ---
> > >
> > > Thank you again for your time and effort in reviewing of paper! We hope our explanations can solve your concern and demonstrate the improved quality of our paper!
> > >
> > > Best Regards,
> > > The Authors

---

### Official Review · Reviewer_oZvA · 2025-03-14

**Overall Recommendation:** 3

**Summary:**

This paper introduces UP-VLA, a vision-language-action model that can understand, generate predicted future images, and plan actions in the embodied environment. It devises a novel VLA training paradigm that unifies policy learning with visual prediction and multi-modal understanding. The results show that the use of future image prediction can significantly improve the precision and visual generalization of policy.

## update after rebuttal:

After reading the results from the reviewer dmnE and FLHB, I was convinced and recognize the contribution "pretraining with mixed multi-modal understanding and future prediction objectives helps the VLA models". So I will change the score to "Weekly accept".

**Claims And Evidence:**

The claims made in the submission supported by clear and convincing evidence

**Essential References Not Discussed:**

All essential references are discussed.

**Experimental Designs Or Analyses:**

Yes, the experiment well designed with good analyses.

**Methods And Evaluation Criteria:**

The proposed methods and evaluation criteria make sense for the problem.

**Other Comments Or Suggestions:**

I think this paper lying a competitive area which is highly care about the metric number.
I think the baseline could be better and stronger e.g.
openvla/octo/openvla-oft
pi0 series
 - https://www.physicalintelligence.company/blog/pi0
 - https://www.physicalintelligence.company/research/fast
VLA + reasoning
 - https://embodied-cot.github.io/

I suggest the author to dig some extra benefits of your investigate, otherwise you have to compare with the sota results. For example, the predicted image is pretty good and could enforce some downstream tasks?

Besides that, I think the framework in this paper is hard to train. May have to carefully desgin the ablation study which is not enough in this paper.

**Other Strengths And Weaknesses:**

The VLA is a big and important community, and the solution of this paper is reasonable.
Experiments are sufficient and solid.

**Questions For Authors:**

refer to the aboves.

**Relation To Broader Scientific Literature:**

This paper contributes for the direction of VLA foundation model training, which is a foundmental and important area. The proposed training strategy and model design are valuable for that community.

**Theoretical Claims:**

No theoretical claims.

---

> ### Author Rebuttal · Authors · 2025-04-01
>
> We sincerely appreciate your time and efforts in reviewing our paper! Based on your review, we added a detailed discussion and additional experiments.
>
> ---
>
> **Q1: About stronger baselines.**
>
> ANS: Thank you for your suggestions! We conduct more experiment on Pi0, OpenVLA, and Octo model. Since these methods only conduct experiments in real world which is hard to replicated, we utilize their open-source code to fine-tune and test them on the Calvin ABC-D (with settings consistent with UP-VLA). The results on the ABC-D task are shown in the table below, where an asterisk* indicates the results we reproduced using the open-source code.
>
> |  Method  |      Type      | Avg. Len ↑ |
> | :------: | :------------: | :--------: |
> | Openvla* |      VLA       |    1.60    |
> |  Octo*   |      VLA       |    0.59    |
> |   pi0*   |      VLA       |    3.63    |
> |  UP-VLA  | Prediction&VLA |    4.08    |
>
> For the pi0-fast method, it primarily improves training efficiency by preprocessing actions. In terms of performance, it is almost identical to pi0. Therefore, we only present the reproduced results of pi0 here.
>
> ---
>
> **Q2: I suggest the author to dig some extra benefits of your investigate, otherwise you have to compare with the sota results. For example, the predicted image is pretty good and could enforce some downstream tasks?**
>
> ANS: Thanks for your insightful suggestions. We provide discussions from the following aspects:
>
> (1) Performance: As you noted, OpenVLA and Pi0-series models leverage powerful VLMs to enhance semantic understanding, similar to our UP-VLA, which employs Showo, a Unified VLM. However, we observed that these methods are good at semantic understanding but lack action precision. The results in Table [Q1] highlight their limitations on the Calvin benchmark. We notice the primary failure mode of these methods in CALVIN stems from imprecise actions(e.g., the robot arm correctly approaches the target object based on the command but fails to grasp it). In contrast, our approach mitigates these shortcomings by integrating visual prediction into the VLA model, leading to more precise actions and superior performance, as shown in Table 1 and Figure 6 of our paper.
>
> (2) Additional benefits of our proposed unified VLA training paradigm: Previous VLA models did not fully exploit the temporal and visual information encoded in video datasets. Our approach introduces a scalable training paradigm that integrates MMU datasets, video datasets, and robot datasets. This allows the robot to benefit from the language comprehension capabilities of VLMs while also excelling in fine-grained manipulation tasks with complex dynamics. As you mentioned, we indeed found that **precise visual prediction can leads to precise actions**, which is a key factor in our method's effectiveness. Some of these precise predictions are visualized in Figure 7. As a result, UP-VLA demonstrates strong performance on both the Calvin benchmark and real-world scenarios.
>
> ---
>
> **Q3: I think the framework in this paper is hard to train.**
>
> ANS: We completely agree that training with multiple objectives across multiple datasets can be more challenging than training with a single objective. Fortunately, several works in the generative modeling field have unified understanding and generation within a single model, such as TransFusion, Show-O, and others. These works provide a valuable foundation for our approach. In this paper, we build UP-VLA upon Show-O, and we have included all the source code in the supplementary material. Feel free to check it for more details on the training process!
>
>
> ---
>
> **Q4: May have to carefully design the ablation study**
>
> ANS: Yes! We have conducted comprehensive ablation studies by systematically removing different components respectively from our original framework to verify the effectiveness of each training objective. As reported in Table 3 of the original paper, eliminating multi-modal understanding pretraining, bridge image prediction pretraining, or image prediction fine-tuning all result in **a clear performance drop**.
>
>
> Also, following Reviewer-dmnE, we conduct extra ablation study on multi-steps. Specifically, we change the length of action chunk with and without visual prediction to validate the effectiveness of our proposed visual prediction objective.
>
> | Ablating Effectiveness of Visual Prediction |   1   |   4   |   7   |  10  |
> | :-----------------------------------------: | :---: | :---: | :---: | :--: |
> |          w/o predict future image           | 1.44 | 1.94 | 2.17 | 2.25 |
> |           w/ predict future image           | 2.42 | 3.72 | 4.00 | 4.08 |
>
> ---
>
> Thank you again for your time and effort in reviewing our work! We hope our clarification can solve all your concerns, and we are always ready to answer any further questions!

---

> > ### Comment · Reviewer_oZvA · 2025-04-02
> >
> > I have read the authors' rebuttal and will keep my score.

---

> > > ### Author Response · Authors · 2025-04-04
> > >
> > > Dear Reviewer oZvA:
> > >
> > > We sincerely appreciate the time and effort you have taken to review our rebuttal. In response to your insightful feedback, we have **carefully incorporated all the requested baseline** comparisons and **highlighted the novel training paradigm** of our approach.
> > >
> > > Could we politely ask if there are any further concerns existed? We are always willing to address any of your further questions. If there are no additional concerns, we sincerely wish you could reconsider your score.
> > >
> > > Thank you once again for your valuable time！
> > >
> > > Best Regards,
> > >
> > > The Authors

---

### Official Review · Reviewer_dmnE · 2025-03-16

**Overall Recommendation:** 4

**Summary:**

This paper presents UP-VLA, a Unified Vision-and-Language Alignment (VLA) model trained with dual objectives: multimodal understanding and future prediction. Building on the foundation of the show-o framework, UP-VLA significantly improves both in-domain performance and generalization to unseen scenarios across real-world and simulation tasks.

**Claims And Evidence:**

The claim regarding the unified VLA paradigm is clear.
However, I personally disagree with the authors' argument about "limiting their ability to capture detailed spatial information and understand physical dynamics."

Firstly, many existing works have already explored similar ideas, including GR-1 cited by the authors.
Secondly, there is substantial evidence [1] showing that even predicting videos does not necessarily lead to a comprehensive understanding of physical dynamics.
Finally, in my opinion, performance improvements are more likely attributed to forcing the model to predict future outcomes, which provides a prior for planning actions before execution. A more detailed discussion on this point is provided below.

[1] How Far is Video Generation from World Model: A Physical Law Perspective

**Essential References Not Discussed:**

[0] How Far is Video Generation from World Model: A Physical Law Perspective;
[1]  VLAS: Vision-Language-Action Model With Speech Instructions For Customized Robot Manipulation, ICLR25
[2] Accelerating Vision-Language-Action Model Integrated with Action Chunking via Parallel Decoding
[3] Fine-Tuning Vision-Language-Action Models: Optimizing Speed and Success
[4] Predictive Inverse Dynamics Models are Scalable Learners for Robotic Manipulation, ICLR 25
[5] Video Prediction Policy: A Generalist Robot Policy with Predictive Visual Representations
[6] Towards Generalist Robot Policies: What Matters in Building Vision-Language-Action Models

**Experimental Designs Or Analyses:**

I think the experimental section is relatively comprehensive, but there are still some issues with missing comparisons.

1. The authors reproduced UP-VLA-RT-2* and phi-w/o-mmu*, but one type is VLA and the other type is Prediction. Therefore, we should first determine whether the performance differences are related to the VLA backbone. To address this, results for both methods on other types should be supplemented.

2. In [1] and [2], they both reproduced VLA-RT-2 and found that simply allowing the model to predict multiple future steps could significantly improve the performance of the backbone model. So, is it possible to understand that as long as the model is tasked with predicting future steps, its performance can skyrocket? A similar conclusion comes from OpenVLA-OFT [3], where simple multi-step reasoning alone brought significant performance improvements. Additionally, the experiments on "w/o prediction" in the authors' Table 3 also demonstrate that predicting the future has a substantial impact on performance improvement. Based on the above, is the core reason for the performance improvement of UP-VLA the ability of the VLA model to predict the future, regardless of the specific prediction method? (This point relates to the Claims and Evidence section.)

3. 3D-VLA is not a good baseline. First, its performance is not very strong to begin with. Additionally, its control mode requires reasoning over many steps at once, which prevents the model from dynamically adjusting its actions based on current observations. Furthermore, prediction errors from earlier steps in the point cloud can accumulate significantly. As a result, it is not an appropriate baseline to demonstrate the advantages of the proposed method.

4. My suggestion is for the authors to compare their method with Seer [4], VPP [5], and RoboVLMs [6]. Both [4] and [5] also focus on Prediction and achieve excellent performance, while [6] falls under the VLA category and also performs very well. From this perspective, I do not see a clear necessity for using UP-VLA, which is crucial for me to understand the contribution of this work.



[1]  VLAS: Vision-Language-Action Model With Speech Instructions For Customized Robot Manipulation, ICLR25
[2] Accelerating Vision-Language-Action Model Integrated with Action Chunking via Parallel Decoding
[3] Fine-Tuning Vision-Language-Action Models: Optimizing Speed and Success
[4] Predictive Inverse Dynamics Models are Scalable Learners for Robotic Manipulation, ICLR 25
[5] Video Prediction Policy: A Generalist Robot Policy with Predictive Visual Representations
[6] Towards Generalist Robot Policies: What Matters in Building Vision-Language-Action Models

**Methods And Evaluation Criteria:**

The proposed method makes sense. This is an attempt to adapt the unified MLLM to VLA, and it also yields some interesting conclusions.

**Other Comments Or Suggestions:**

Please refer to the experiments.

**Other Strengths And Weaknesses:**

N/A

**Questions For Authors:**

N/A

**Relation To Broader Scientific Literature:**

I think combining VLA with a world model has potential, but based on the author's current explanation, I don't see it yet.

**Theoretical Claims:**

I have checked the method, but since there is hardly much theoretical proof, I won't discuss it further here.

---

> ### Author Rebuttal · Authors · 2025-04-01
>
> We sincerely appreciate your time and efforts in reviewing our paper! Based on your review, we added a detailed discussion and additional experiments.
>
> ---
>
> **Q1: predicting videos does not necessarily lead to a comprehensive understanding of physical dynamics**
>
> ANS: Whether video models can truly understand physical dynamics remains an open research question, with arguments on both sides [1]. In our experiments, we found that training with a video prediction objective enables UP-VLA models to generate physically consistent future frames. Moreover, we think that better understanding can be reflected in better decision-making and control capabilities. As shown in Table [Q2], our ablation studies demonstrate that video prediction objective consistently improves robot control performance in all settings. We hope this clarification can solve your concern!
>
> [1] https://openai.com/index/video-generation-models-as-world-simulators
>
>
> ---
>
> **Q2: Maybe performance improvements are more likely attributed to forcing the model to predict multiple future outcomes (future steps), as shown in VLAS, OpenVLA-OFT**
>
> ANS: Thank you for your insightful question! We agree that multi-step prediction can benefit action learning, which we adopted as the default in our experiments. However, beyond multi-step prediction, our primary goal was to show that **predicting future images** (RGB modality) also improves action learning by facilitating information transfer from video datasets (image modality) to action learning (action modality).
>
> To further illustrate the impact of multi-step action prediction and RGB image prediction, we conducted a detailed ablation study along these two axes. Our results show that increasing the number of predicted action steps and incorporating future image predictions both contribute to performance improvements. We hope this ablation further clarifies the effectiveness of UP-VLA, which integrates image prediction objectives into VLA model.
>
> | Length of predicted actions |   1   |   4   |   7   |  10  |
> | :-----------------------------------------: | :---: | :---: | :---: | :--: |
> |          w/o predict future image           | 1.44 | 1.94 | 2.17 | 2.25 |
> |           w/ predict future image           | 2.42 | 3.72 | 4.00 | 4.08 |
>
>
> ---
>
> **Q3:The authors reproduced UP-VLA-RT-2 and phi-w/o-mmu, but one type is VLA and the other type is Prediction. Therefore, we should first determine whether the performance differences are related to the VLA backbone. To address this, results for both methods on other types should be implemented.**
>
> ANS: Thank you for your comments. The Show-o model is fine-tuned from the Phi LLM on multimodal understanding datasets. Since we aimed to fully remove the influence of multi-model understanding tasks, we choose to start from the original Phi model in the phi-w/o-mmu ablation.
> Following your suggestion, we conducted a comprehensive comparison to better highlight the advantages of UP-VLA across different backbones. These experiments further demonstrate that both video prediction and multimodal understanding contribute to improved action learning.
>
> |   Method   | Backbone | Avg. Len ↑ |
> | :--------: | :------: | :--------: |
> |    VLA     |  Phi1.5  |    0.79    |
> | Prediction |  Phi1.5  |    3.13    |
> |    VLA     |  Show-o  |    1.44    |
> | Prediction |  Show-o  |    3.99    |
>
> ---
>
> **Q4: Compare with Seer, VPP, and RoboVLMs. Both Seer and VPP also focus on Prediction and achieve excellent performance. RoboVLM falls under the VLA category and also performs very well.**
>
> ANS: Thank you for your insightful comments! We would like to argue that UP-VLA has significant difference on learning paradigm with these previous works. Seer and VPP leverage internet video datasets to aid robot learning, while RoboVLM utilizes a pretrained VLM. Each of these methods relies on a single type of dataset—either video datasets or multimodal understanding datasets. To the best of our knowledge, UP-VLA is the first model to leverage both image prediction (video) and multimodal understanding (MMU) datasets for embodied decision-making. Furthermore, our ablation studies confirm that both types of data significantly contribute to the final performance.
>
> We also note that Seer, VPP, RoboVLM, and UP-VLA all achieve an average task completion length exceeding 4.0 on the CALVIN benchmark. However, these three works adopt distinct architectures from UP-VLA and are concurrent with UP-VLA. To the best of our knowledge, UP-VLA achieves the best performance on CALVIN ABC benchmark among none-concurrent works.
>
>
> ---
>
> We hope our clarifications address your concerns and demonstrate the improved quality of our paper! Please feel free to reach out with any further questions.
> Thank you again for your valuable time!

---

> > ### Comment · Reviewer_dmnE · 2025-04-04
> >
> > I appreciate the additional experiments conducted by the authors, as training these models within a short period is indeed not an easy task. I believe the authors have understood my comments on the paper, and the added experiments align with my expectations: the method of predicting future goals essentially forces the VLA to engage in planning, thereby improving the success rate of task execution.
> >
> > I think the contribution of this paper lies in the novel application of a unified MLLM architecture to robotic tasks, which is undoubtedly significant. Since the authors have addressed my concerns, I am inclined to recommend acceptance of this paper, as it could provide valuable insights for the design of future VLA models.
> >
> > Of course, regarding the concerns about writing details, I hope the authors can address them before the final version is submitted.

---

> > > ### Author Response · Authors · 2025-04-04
> > >
> > > Dear Reviewer dmnE:
> > >
> > > We are thrilled to receive your feedback! Your support for our work is truly appreciated. We will incorporate all the new experiments into the final version and continue refining it to meet the highest standards!
> > >
> > > Best Regards,
> > >
> > > The Authors

---

### Decision · Program_Chairs · 2025-05-01

**Decision:**

Accept (poster)

**Comment:**

This paper proposes UP-VLA, a unified VLA model that integrates multi-modal understanding and future image prediction to improve embodied decision-making. The authors present extensive empirical studies across simulation and real-world tasks, and provide additional ablations and baseline comparisons during the rebuttal phase.


Reviewer dmnE and FLHB value the direction of unifying prediction and understanding in VLA and believe UP-VLA presents a compelling method. While acknowledging the limited novelty, the reviewer finds the practical challenges of such integration non-trivial and supports acceptance due to the model’s inspiration and empirical value to the field.
Reviewer dQTi initially raised concerns about inference efficiency, lack of SOTA VLA baselines (e.g., OpenVLA, pi0), and missing implementation details. After the rebuttal, the reviewer was satisfied with the additional experimental support.
Reviewer oZvA raised concerns over missing stronger baselines, limited insights from the visual prediction component, and insufficient ablation. The reviewer decided to maintain the original score after rebuttal.

In summary, while all reviewers agree that the technical novelty is somewhat modest, as UP-VLA builds largely on the existing Show-o framework, and that the claim of being “unified” may be overstated given its reliance on the Bridge dataset, the paper nonetheless addresses a timely and important problem. VLA research is still in its early stages, and the exploration of unified pretraining strategies is a meaningful contribution that could inform future developments in the field.